# GREEDY PIG: ADAPTIVE INTEGRATED GRADIENTS

## ABSTRACT

Deep learning has become the standard approach for most machine learning tasks. While its impact is undeniable, interpreting the predictions of deep learning models from a human perspective remains a challenge. In contrast to model training, model interpretability is harder to quantify and pose as an explicit optimization problem. Inspired by the AUC softmax information curve (AUC SIC) metric for evaluating feature attribution methods, we propose a unified discrete optimization framework for feature attribution and feature selection based on subset selection. This leads to a natural *adaptive* generalization of the path integrated gradients (PIG) method for feature attribution, which we call Greedy PIG. We demonstrate the success of Greedy PIG on a wide variety of tasks, including image feature attribution, graph compression/explanation, and post-hoc feature selection on tabular data. Our results show that introducing adaptivity is a powerful and versatile method for making attribution methods more powerful.

## 1 INTRODUCTION

Deep learning sets state-of-the-art on a wide variety of machine learning tasks, including image recognition, natural language understanding, large-scale recommender systems, and generative models (Graves et al., 2013; He et al., 2016; Krizhevsky et al., 2017; Bubeck et al., 2023). Deep learning models, however, are often opaque and hard to interpret. There is no native procedure for *explaining* model decisions to humans, and explainability is essential when models make decisions that directly influence people's lives, *e.g.*, in healthcare, epidemiology, legal, and education (Ahmad et al., 2018; Wiens & Shenoy, 2018; Chen et al., 2021a; Abebe et al., 2022; Liu et al., 2023).

We are interested in two routines: (i) *feature attribution* and (ii) *feature selection*. Given an input example, feature attribution techniques offer an explanation of the model's decision by assigning an *attribution score* to each input feature. These scores can then be directly rendered in the input space, *e.g.*, as a heatmap highlighting visual saliency (Sundararajan et al., 2017), offering an explanation of model decisions that is human-interpretable. On the other hand, feature selection methods find the set of most-informative features for a machine learning model across many examples to optimize its prediction quality. While the goal of both feature attribution and feature selection is to find the most impactful features, they are inherently different tasks with a disjoint set of approaches and literature. Concretely, feature attribution considers one example per invocation, while feature selection considers an entire dataset. For literature surveys, see (Zhang et al., 2021) for feature attribution and interpretability see and (Li et al., 2017) for feature selection.

Our main contributions are as follows:

- We consider the problem of feature attribution from first principles and, inspired by evaluation metrics in the literature (Kapishnikov et al., 2019), we propose a new formulation of feature attribution as an explicit subset selection problem.
- Using this subset selection formulation, we ascribe the main shortcomings of the *path integrated gradient* (PIG) algorithms to their limited ability to handle feature correlations. This motivates us to introduce adaptivity to these algorithms, which is a natural way to account for correlations, leading to a new algorithm that we call Greedy PIG.
- We leverage the generality of the subset selection formulation to propose different objectives for feature attribution based on model loss that can better capture the model's behavior, as well as a unification between feature attribution and feature selection formulations.
- We evaluate the performance of Greedy PIG on a variety of experimental tasks, including feature attribution, graph neural network (GNN) compression, and post-hoc feature selection.

Our results show that adaptivity is a powerful ingredient for improving the quality of deep learning model interpretability methods.

## 2 Preliminaries & notation

### 2.1 Path integrated gradients (PIG)

Let $f(\cdot; \boldsymbol{\theta})$ be a pre-trained neural network with parameters $\boldsymbol{\theta}$. Our work assumes that $f$ is pre-trained and therefore, we often omit $\boldsymbol{\theta}$ for brevity. Suppose $\boldsymbol{x} \in \mathbb{R}^n$ is an input example. The *path integrated gradients* (PIG) method of Sundararajan et al. (2017) attributes the decision of $f$ given $\boldsymbol{x}$, by considering a *line path* from the "baseline"[1] example $\boldsymbol{x}^{(0)}$ to $\boldsymbol{x}$:

$$\gamma : [0,1] \to \mathbb{R}^n \quad \text{with} \quad \gamma(t; \boldsymbol{x}^{(0)}, \boldsymbol{x}) = \boldsymbol{x}^{(0)} + t(\boldsymbol{x} - \boldsymbol{x}^{(0)}) \quad \text{for} \quad t \in [0,1]. \tag{1}$$

Then, the top features $i \in [n]$ maximizing the following (weighted) integral are chosen:

$$\underset{i \in [n]}{\arg\,\text{sort}} \left[ (\boldsymbol{x}_{[i]} - \boldsymbol{x}_{[i]}^{(0)}) \int_0^1 \frac{\partial}{\partial \boldsymbol{x}_{[i]}} f(\gamma(t)) dt \right]. \tag{2}$$

Our work addresses a weakness of PIG, which is that it is a one-shot algorithm. Specifically, an invocation to PIG computes (in-parallel) Eq. 1 and chooses (at-once) the indices $\subseteq [n]$ maximizing Eq. 2.

### 2.2 Subset selection

Subset selection is a family of discrete optimization problems, with goal of selecting subset $S \subseteq [n] := \{1, 2, \ldots, n\}$ that maximizes a set function $G : 2^{[n]} \to \mathbb{R}$ subject to a cardinality constraint $k$:

$$S^* = \underset{S \subseteq [n], |S| \leq k}{\arg\max} \ G(S). \tag{3}$$

Prominent examples of subset selection in machine learning include data summarization (Kleindessner et al., 2019), feature selection (Cai et al., 2018; Elenberg et al., 2018; Bateni et al., 2019; Chen et al., 2021b; Yasuda et al., 2023), and submodular optimization (Krause & Golovin, 2014; Mirrokni & Zadimoghaddam, 2015; Fahrbach et al., 2019b;a).

Throughout this work, we let $\mathbf{1}$ denote $n$-dimensional all-one vector, $\mathbf{1}_S \in \{0,1\}^n$ be the indicator vector of $S$, and let $\boldsymbol{x}_S = \mathbf{1}_S \odot \boldsymbol{x} \in \mathbb{R}^n$ where $\odot$ denotes Hadamard product.

## 3 Our method

### 3.1 Motivation: Discrete optimization for feature attribution and selection

The general subset selection framework (§2.2) allows recovering several tasks. For (i) feature attribution, the *softmax information curve (SIC)* of Kapishnikov et al. (2019) can be recovered from (Eq. 3) by setting $G(S)$ to the softmax output of a target class (see Eq. 4). For (ii) feature selection, we can set $G(S)$ to maximum log likelihood achievable by a model trained on a subset of features $S$. Finally, if one sets $G(S)$ to be the maximum model output perturbation achieved by only changing the values of features in $S$, one recovers a task of finding *counterfactual explanations*.

The generality of framework (§2.2) encourages us to pose the tasks (i) and (ii) as instances. Consequently, we inherit algorithms and intuitions from the subset selection and submodular optimization literature. The area of submodular optimization has a vast literature with different theoretical analyses and algorithms based on the specific properties of the set function. For example, for maximization of weakly submodular functions, it is known that the greedy subset selection algorithm achieves a constant approximation ratio. In general, determining the right realistic assumptions on the set function is a major open problem in the area. As such, it is not yet clear which of these assumptions are realistic for each use case. We believe this is a very important question for future work.

---

[1]Sundararajan et al. (2017) uses a black-image and Kapishnikov et al. (2019) averages the integral over black-and white-images. Random noise baselines have also been considered.

## 3.2 Instantiations

**Multi-class attribution.** Suppose that classifier $f$ is the softmax output of a multi-class neural network, then the softmax information curve of Kapishnikov et al. (2019) can be written in (§4.1; Eq. 3) as:

$$G^{\text{ATTRIBUTIONSOFTMAX}}(S) = f_{c^*}(\boldsymbol{x}_S), \quad \text{with} \quad c^* = \arg\max_j f_j(\boldsymbol{x}). \tag{4}$$

Maximizing $S$ in Eq. 4 chooses feature subset $S$ that maintains the model's decision, as compared to the fully-observed example $\boldsymbol{x}$. Crucially, however, the top class $c^*$ doesn't tell the whole story, since the input might be ambiguous or the classifier might be uncertain. It is then natural that one would want to capture the behavior of $f$ on *all* classes, rather than just the top one. A natural alternative is to re-use the objective ($\ell$) that the model was trained with, and incorporate it into a subset selection problem. Specifically, we define the set function:

$$G^{\text{ATTRIBUTIONKL}}(S) = \ell(f(\boldsymbol{x}), f(\boldsymbol{x}_S)) = f(\boldsymbol{x}) \log f(\boldsymbol{x}_S), \tag{5}$$

where, for classification, $\ell$ is the log-likelihood. This quantifies the similarity of probability distribution $f(\boldsymbol{x}_S)$, i.e., considering a subset of features, with the distribution $f(\boldsymbol{x})$, i.e., considering all features. Maximizing $G(S)$ under a cardinality constraint is then equivalent to seeking a small number of features that capture the multiclass behavior of the model on a fixed input example.

**Feature selection.** Unlike feature attribution, outputting attributions for *a single example*, in feature selection, the goal is to select a *global set* of features for *all dataset examples*, that maximize the predictive capability of the model. Feature selection can be formulated by defining

$$G^{\text{FEATSELECT}}(S) = \max_{\boldsymbol{\theta}} \ell(\boldsymbol{y}, f(\boldsymbol{X}_S; \boldsymbol{\theta})), \tag{6}$$

*i.e.*, the maximum log likelihood obtainable by training the model $f$ on the whole dataset $\boldsymbol{X}$, given only the features in $S$. Such formulation of feature selection as subset selection has been studied extensively, e.g., Liberty & Sviridenko (2017); Elenberg et al. (2018); Bateni et al. (2019); Chen et al. (2021b); Yasuda et al. (2023); Axiotis & Yasuda (2023). To avoid re-training $f$, we consider a simpler problem of *post-hoc feature selection*, with $G^{\text{FEATSELECTPH}}(S) = \ell(\boldsymbol{y}, f(\boldsymbol{X}_S; \boldsymbol{\theta}))$ where the model parameters $\boldsymbol{\theta}$ are pre-trained on all features. We leave further investigation of (6) to future work. The post-hoc formulation has the advantage of making the objective differentiable, and thus directly amenable to gradient-based approaches, while also not requiring access to the training process of $f$, which is seen as a black box. This can be particularly useful for applications with limited resources or engineering constraints that discourage running or modifying the training pipeline. As we will show in §5.3, high quality feature selection can be performed even in this restricted setting.

## 3.3 Continuous extension

To make the problem (Eq. 3) amenable to continuous optimization methods, we rely on a continuous extension $g : [0,1]^n \to \mathbb{R}$ with $G(S) := g(\boldsymbol{1}_S)$. The extension can be derived by replacing the invocation of $f$ on $\boldsymbol{x}_S$ or $\boldsymbol{X}_S$ in Eq. 4–6 by an invocation on a *path*, similar to Eq. 1, but with domain equal to the $n$-hypercube:

$$g : [0,1]^n \to \mathbb{R}^n \quad \text{with} \quad g(\mathbf{s}; \boldsymbol{x}^{(0)}, \boldsymbol{x}) = \boldsymbol{x}^{(0)} + \mathbf{s} \odot (\boldsymbol{x} - \boldsymbol{x}^{(0)}) \quad \text{for} \quad \mathbf{s} \in [0,1]^n. \tag{7}$$

This makes it easy to adapt feature attribution methods from the literature to our framework and is a lightweight assumption since loss functions in deep learning are continuous by definition. In particular, the integrated gradients algorithm of Sundararajan et al. (2017) can simply be stated as computing the vector of feature scores given by $\int_{t=0}^{1} \nabla g(t\boldsymbol{1}) dt$.

## 3.4 Greedy path integrated gradients (Greedy PIG)

Given the framework in §3.1, we now move to identify and improve the weaknesses of the integrated gradients algorithm. In contrast to previous works that focus on reducing the amount of "noise" perceived by humans in the attribution result,[2] we identify a different weakness of PIG—it is *sensitive to feature correlations*. In §4.1, we show that even in simple linear regression settings, PIG exhibits pathologies such as outputting attribution scores that are negative, or whose ordering can be manipulated via feature replication. This leads us to introduce adaptivity to the algorithm. We are motivated by the greedy algorithm for subset selection, which is known to provably tackle submodular optimization problems, even when there are strong correlations between features.

---

[2]In fact, as noticed in Smilkov et al. (2017), a noisy attribution output is not necessarily bad, since our goal is to *explain model predictions* and not produce a human-interpretable understanding of the *data*, which is an important but different task. Neural net predictions do not have to conform to human understanding of content.

**Why Greedy captures correlations.**    A remarkable property of the greedy algorithm is that its adaptivity property can automatically deal with correlations. Consider a set function $G$ and three elements $i, j, k$. Suppose that greedy initially selects element $i$. Then, it will recurse on the residual set function $G'(S) = G(S \cup \{i\})$. Thus, in the next phases, correlations between $i$ and the remaining elements are eliminated, since $i$ is fixed (selected). In other words, since greedy conditions on the already selected variables, this eliminates their correlations with the rest of the unselected variables.

**Greedy PIG.**    Inspired by the adaptivity complexity of submodular maximization (Balkanski & Singer, 2018), where it has been shown that $\Omega(\log n / \log \log n)$ adaptive rounds of evaluating $G$ are needed to achieve provably-better solution quality than one-shot algorithms (such as integrated gradients), we propose a generalization of integrated gradients with multiple rounds. We call this algorithm Greedy PIG since it greedily selects the top-attribution features computed by integrated gradients in each round, *i.e.* the set of $S \subseteq [n]$ maximizing the $\arg$ sort of Eq. 2, and hardcodes their corresponding entries in $\mathbf{s}$ to 1.

### 3.5    The greedy PIG algorithm

Our idea is to iteratively compute the top features as attributed by integrated gradients, and select these features by always including them in future runs of integrated gradients (and only varying the rest of the variables along the integration path). We present this algorithm in detail in Algorithm 1.

---

**Algorithm 1** Greedy PIG (path integrated gradients)

---

1: **Input:** access to a gradient oracle for $g : [0,1]^n \to \mathbb{R}$
2: **Input:** number of rounds $R$, number of selections per round $z$
3: Initialize $S \leftarrow \varnothing$                                                 ▷ Selected set of features
4: Initialize $\boldsymbol{a} \leftarrow \boldsymbol{0}$                                 ▷ Vector of attributions
5: **for** $r = 1$ to $R$ **do**
6:      Set $\hat{\boldsymbol{a}} \leftarrow \int_{t=0}^{1} \nabla_{\bar{S}} g(\mathbf{1}_S + t\mathbf{1}_{\bar{S}}) dt$
7:      Set $\hat{S} \leftarrow \text{top}_z(\hat{\boldsymbol{a}})$                        ▷ Top-$z$ largest entries of $\hat{\boldsymbol{a}}$
8:      Update $\boldsymbol{a} \leftarrow \boldsymbol{a} + \hat{\boldsymbol{a}}_{\hat{S}}$
9:      Update $S \leftarrow S \cup \hat{S}$
      **return** $\boldsymbol{a}, S$

---

In practice, Greedy PIG can be implemented by repeated invocation of PIG (Eq. 1&2). The impact of $g$ (Eq. 7) can be realized by iteratively updating:

$$\hat{S} \leftarrow \text{top indices of Eq. 2 with } \quad \gamma(.; \boldsymbol{x}^{(j+1)}, \boldsymbol{x}) \quad \text{with} \quad \boldsymbol{x}^{(j+1)} \leftarrow \boldsymbol{x}_{\hat{S}} + \boldsymbol{x}^{(j)}_{\bar{\hat{S}}} \tag{8}$$

Specifically, Greedy PIG is related to a continuous version of Greedy. In the continuous setting, selecting an element $i$ is equivalent to fixing the $i$-th element of the baseline to its final value. The $i$-th element now does not vary and its correlations with the remaining elements are eliminated. At a high level, our approach has similarities to (Kapishnikov et al., 2021), which adaptively computes the trajectory of the PIG path based on gradient computations.

Similar to previous work, the integral in Algorithm 1 on Line 6 can be estimated by discretizing it into a number of steps. A simpler approach that we found to be competitive and more frugal is to approximate the integral by a single gradient $\nabla_{\bar{S}} g(\mathbf{1}_S)$, which we call the *Sequential Gradient (SG)* algorithm. It should be noted that, in contrast to integrated gradients, Algorithm 1 empirically does not return negative attributions.

## 4    Analysis

### 4.1    Attribution as subset selection

The integrated gradients method is widely used because of its simplicity and usefulness. Since its inception, it was known to satisfy two axioms that Sundararajan et al. (2017) called sensitivity and implementation invariance, as well as the convenient property that the sum of attributions is equal to the function value. However, there is still no deep theoretical understanding of when this method (or any attribution method, for that matter) succeeds or fails, or even how success is defined.

Previous works have studied a variety of failure cases. In fact, it is well-observed that the attributions on image datasets are often "grainy," a property deemed undesirable from a human standpoint (although as stated before, it is not necessarily bad as long as the attributions are highly predictive of the model's output). In addition, there will almost always exist some negative attribution scores, whose magnitude can be even higher than that of positive attribution scores. There have been efforts to mitigate these effects for computer vision tasks, most notably by ascribing them to noise (Smilkov et al., 2017; Adebayo et al., 2018), region grouping (Ancona et al., 2017; Kapishnikov et al., 2019), or ascribing them to the magnitude of the gradient (Kapishnikov et al., 2021).

In this work, we take a different approach and start from the basic question:

*What are attribution scores supposed to compute?*

An important step towards answering this question was done in (Kapishnikov et al., 2019), which defined the *softmax information curve (SIC)* for image classification. In short, the idea is that if we only keep $k$ features with the top attributions and drop the rest of the features, the softmax output of the predicted class should be as large as possible. This is because then, the model prediction can be distilled down to a small number of features.

Based on the above intuition, we define the problem in a more general form:

**Definition 4.1** (Attribution as subset selection). Given a set function $G : 2^{[n]} \to \mathbb{R}_{>0}$, a value $k \in [n]$ and a permutation $\boldsymbol{r} = (r_1, r_2, \ldots, r_n)$ of $[n]$, the *attribution quality* of $\boldsymbol{r}$ at level $k$ is defined as $G(\{r_1, r_2, \ldots, r_k\})/\max_{S \subseteq [n]:|S|=k} G(S)$. We define the attribution quality of $\boldsymbol{r}$ (without reference to $k$) as the area under the curve defined by the points $(k, G(\{r_1, r_2, \ldots, r_k\}))$ for all $k \in [n]$.

The key advantage of this approach is that we are able to pose attribution as an explicit optimization problem, and hence we are able to compare, evaluate, and analyze the quality of attribution methods. In addition, the formulation in Definition 4.1 gives us flexibility in picking the objective $G$ to be maximized. In fact, in §5.1 we propose setting $G$ to be the log likelihood (*i.e.*, negative cross entropy loss) instead of the top softmax score. In §5.3, we will see how this formulation allows us to tie attribution together with feature selection.

**Note.** As Definition 4.1 shows, we are interested in the problem of attributing predictions to a *small* number of features. For example, when faced with redundant features, we wish to pinpoint only a small number of them. This does not capture applications in which the goal is to find the attribution for *all* features, where the goal would be to assign equal attribution score to all redundant features.

## 4.2 CORRELATIONS AND ADAPTIVITY

Given the problem formulation in Definition 4.1, we are ready to study the strengths and weaknesses of attribution methods. First, it is important to define the notion of *marginal gain*, which is a central notion in subset selection problems.

**Definition 4.2** (Marginal gain). Given a set function $G : 2^{[n]} \to \mathbb{R}_{\geq 0}$, the marginal gain of the $i$-th element at $S \subseteq [n]$ is defined by $G(S \cup \{i\}) - G(S)$. When $S$ is omitted, we simply define the marginal gain as $G(\{i\}) - G(\varnothing)$.

Marginal gains are important because they are closely related to the well-studied greedy algorithm for subset selection (Nemhauser et al., 1978), and are crucial for its analysis. Therefore, a natural question arises: *Are the outputs of attribution methods correlated with the marginal gains?*

For this, let us assume that the set function $G$ is induced by a continuous relaxation $g : [0, 1]^n \to \mathbb{R}$. We show that the integrated gradient attributions approximate the marginal gains up to an additive term that depends on the second-order behavior of $g$, and corresponds to the amount of correlation between variables. In fact, when the variables are uncorrelated, integrated gradient attributions are equal to the marginal gains.

**Lemma 4.3** (PIG vs marginal gain). *Let $H(\boldsymbol{w})$ be the Hessian of a twice continuously differentiable function $g : [0, 1]^n \to \mathbb{R}$ at $\boldsymbol{w}$. Then, for all $i \in [n]$,*

$$\left| \int_{t=0}^{1} \nabla_i g(t\boldsymbol{1}) dt - (g(\boldsymbol{1}_i) - g(\boldsymbol{0})) \right| \leq \frac{1}{2} \max_{\boldsymbol{w} \in [0,1]^n, i \in [n]} \left| \sum_{j \neq i} H_{ij}(\boldsymbol{w}) \right|,$$

*where $\boldsymbol{1}_i$ is the $i$-th standard basis vector of $\mathbb{R}^n$.*

Lemma 4.3 tells us that the quality of how well the integrated gradient attribution scores approximate the marginal gains is strongly connected to correlations between input variables. Indeed, if these correlations are too strong, the attributions can diverge in magnitude and sign, even in simple settings. This motivates combining integrated gradients with adaptivity (e.g. the Greedy algorithm), since adaptivity naturally takes correlations into account. The Greedy PIG algorithm, which we will define in §3.5, is a natural combination of integrated gradients and Greedy.

We now look at a more concrete failure case of integrated gradients in Lemma 4.4, that can arise because of feature redundancy, and how adaptivity can help overcome this issue.

**Lemma 4.4** (Feature redundancy). *Consider a continuous set function $g : [0,1]^n \to \mathbb{R}$ and $t$ redundant features numbered $1, 2, \ldots, t$, or in other words for any $\boldsymbol{w} \in [0,1]^n$ we have $g((w_1, \ldots, w_n)) = h((\max\{w_1, \ldots, w_t\}, w_{t+1}, \ldots, w_n))$ for some $h : [0,1]^{n-t+1} \to \mathbb{R}$. Then, the integrated gradients algorithm with a baseline of $\mathbf{0}$ will assign equal attribution score to features $1$ through $t$.*

It follows from Lemma 4.4 that it is possible to replicate the top-attributed feature multiple times such that the top attributions are all placed on redundant copies of the same feature, therefore missing features $t + 1$ through $n$. This behavior arises because integrated gradients is a one-shot algorithm, but it can be remedied by introducing adaptivity. Specifically, if we select *one* of the features with top attributions, e.g., feature $1$ and then re-run integrated gradients with an updated baseline $(1, 0, \ldots, 0)^\top$, then the new attribution scores of the redundant features $2, \ldots, t$ will be $0$. Therefore, the remaining $k - 1$ top attributions will be placed on the remaining features $t + 1, \ldots, n$.

## 5 EXPERIMENTAL EVALUATION

In this section, we evaluate the effect of the adaptivity introduced by Greedy PIG. A comparison with all attribution methods is out of scope, mainly because our proposed modification can be easily adapted to other algorithms (e.g., SmoothGrad), and so, even though we provide comparisons with different popular methods, we mostly concentrate on integrated gradients. We defer an evaluation of adaptive generalizations of other one-shot algorithms to future work.

### 5.1 FEATURE ATTRIBUTION FOR IMAGE RECOGNITION

**Experimental setup.** We use the MobileNetV2 neural network (Sandler et al., 2018) pretrained on ImageNet, and compare different feature attribution methods on individual Imagenet examples. We use the all-zero baseline for integrated gradients, which, because of the data preprocessing pipeline of MobileNetV2, corresponds to an all-gray image. To ensure that our results hold across models, we present additional results on ResNet50 in the appendix.

**Top-class attribution.** Kapishnikov et al. (2019) introduced the softmax information curve for image classification, which plots the output softmax probability of the target class using only the top attributions, as a function of the compressed image size. Since we are interested in a general method, we instead plot the output softmax as a function of the number of top selected features. To perform attribution based on our framework, we let $c^*$ be the class output by the model $f(\boldsymbol{x}; \boldsymbol{\theta})$ on input $\boldsymbol{x}$ and with parameters $\boldsymbol{\theta}$, and then define $g^{\text{top1}}(\mathbf{s}) := f_{c^*}(\boldsymbol{x} \odot \mathbf{s}; \boldsymbol{\theta}))$, where $\mathbf{s} \in [0,1]^n$ and $\odot$ denotes the Hadamard product. In other words, $g^{\text{top1}}$ gives the softmax output of the (fixed) class predicted by the model, after re-weighting input features by $\mathbf{s}$. We can now run gradient-based attribution algorithms on $g$. The results can be found in Figure 1 and Figure 7.

**Loss attribution.** Even though the top class attribution methodology can explain what are the most important features that influence the model's top predicted class, it might fail to capture other aspects of the model's behavior. For example, if the model has low certainty in its prediction, the softmax output of the top class will be low. In fact, in such cases the attribution method can select a number of features that give a much more confident output. However, it can instead be useful to capture the outputs of *all classes*, and not just the top one. More generally, we can use arbitrary loss functions to attribute certain aspects of the model's behavior to the features.

Specifically, instead of asking which features are most responsible for a specific classification output, we ask: Which features are most responsible for the *distribution* on output classes? In other words, how close is the model's multiclass output vector when fed a subset of features versus when fed all the features? In order to answer this question, we use the loss function on which the model is trained. For multiclass classification, this is usually set to using the negative cross entropy loss $\ell(\bar{\boldsymbol{y}}, \boldsymbol{y}) = \langle \bar{\boldsymbol{y}}, \log \boldsymbol{y} \rangle$ to define a set function $G(S) = \ell(f(\boldsymbol{x}), f(\boldsymbol{x}_S))$ and a corresponding

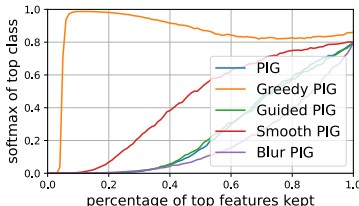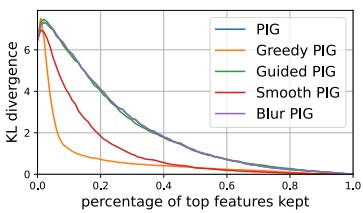

| Algorithm | softmax AUC ($\uparrow$) | KL divergence AUC ($\downarrow$) |
|---|---|---|
| Integrated Gradients (Sundararajan et al., 2017) | 0.2639 | 2.0812 |
| Smooth PIG (Smilkov et al., 2017) | 0.4433 | 1.1366 |
| Blur PIG (Xu et al., 2020) | 0.1903 | 2.0812 |
| Guided PIG (Kapishnikov et al., 2021) | 0.2623 | 2.0644 |
| Greedy PIG | **0.8486** | **0.6655** |

Figure 1: Evaluating the attribution performance across 1466 examples randomly drawn from Imagenet. The Softmax Information Curve (SIC) plots the softmax output of the top class as a function of the number of features kept (there are $224 \times 224 \times 3$ features total for these RGB images). The Loss Information Curve (LIC) similarly plots the KL divergence from the output probabilities on the features kept, to the output probabilities when all features are kept. To compute these results, we run integrated gradients and guided integrated gradients with 2000 steps and Greedy PIG with 100 rounds, 20 steps each. Then, for each algorithm and number of features kept, we compute and plot the median across all examples, as in Kapishnikov et al. (2019). Left: The Softmax Information Curve (higher is better). Right: The Loss Information Curve (lower is better).

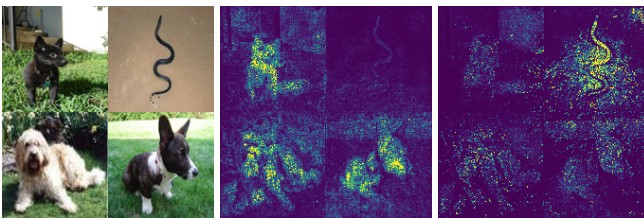

(a) Top 15000 attributions for class "sea snake". Left: input, Middle: integrated gradients, Right: Greedy PIG.

| Algorithm | pointing accuracy ($\uparrow$) |
|---|---|
| Integrated Gradients (Sundararajan et al., 2017) | 0.66 |
| Greedy PIG | **0.83** |

(b) Pointing accuracy aggregated over 25 $2x2$ grids generated from examples that had the highest prediction confidence from 1500 examples randomly drawn from the validation set of ImageNet.

Figure 2: Results from the pointing game Bohle et al. (2021) that is used for sanity checking image attribution methods. We generate $2x2$ grids of the highest prediction confidence images, and obtain the attribution results for each class. For each (example, class) pair, we count it as a positive if the majority of the top 15000 attributions are on the quadrant associated with that class. We measure pointing accuracy as the fraction of such positives.

continuous extension $g^{\text{logloss}}(\mathbf{s}) = \ell(f(\boldsymbol{x}), f(\boldsymbol{x} \odot \mathbf{s}))$. Maximizing $G$ is equivalent to maximizing $\ell(f(\boldsymbol{x}), f(\boldsymbol{x}_S))$. Fortunately, gradient-based attribution algorithms can be easily extended to arbitrary objective functions. The results can be found in Figure 1 and Figure 8.

## 5.2 EDGE-ATTRIBUTION TO COMPRESS GRAPHS & INTERPRET GRAPH NEURAL NETWORKS

We use our method for *graph compression* with pre-trained GCN of Kipf & Welling (2017). We use a three-layer GCN, as:  $\text{GCN}(\mathcal{E}; \theta) = \textbf{softmax}\left(\widehat{A_{\mathcal{E}}}\sigma\left(\widehat{A_{\mathcal{E}}}\sigma\left(\widehat{A_{\mathcal{E}}}\mathbf{X}\theta^{(1)}\right)\theta^{(2)}\right)\theta^{(3)}\right)$, where $\mathbf{X} \in \mathbb{R}^{m \times n}$ contains features of all $m$ nodes, $\mathcal{E} \subseteq [m] \times [m]$ denotes (undirected) edges with a corresponding sparse (symmetric) adjacency matrix with $(A_{\mathcal{E}})_{ij} = 1$ iff in $(i, j) \in \mathcal{E}$; $\widehat{\cdot}$ is the

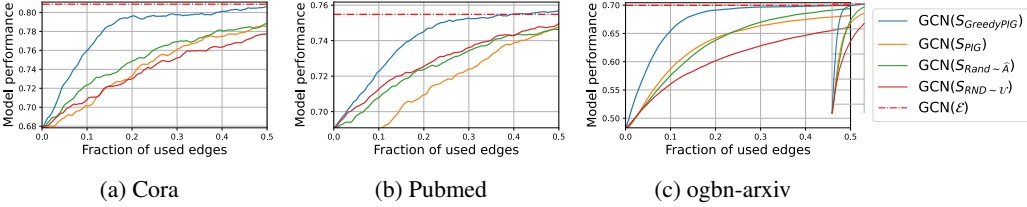

|   (a) Cora   |   (b) Pubmed   |   (c) ogbn-arxiv   |

Figure 3: Pre-trained accuracy GCN with inference using edge-subset selected by various methods. Solid lines plot accuracy vs. ratio of selected edges. Dashed line shows accuracy with all edges.

symmetric-normalization with self-connections added, i.e., $\widehat{A} = (D + \mathbf{I}_m)^{-\frac{1}{2}}(A + \mathbf{I}_m)(D + \mathbf{I}_m)^{-\frac{1}{2}}$; diagonal degree matrix $D = \mathrm{diag}(\mathbf{1}_m^\top A)$; and $\sigma$ denotes element-wise non-linearity such as ReLU.

Given GCN pre-trained for node classification (Kipf & Welling, 2017; Abu-El-Haija et al., 2019; Hamilton et al., 2017), we are interested in inferring node labels using only a subset of the edges. The edge subset can be chosen uniformly at random (Fig. 3: $S_{\mathrm{RND}\sim\mathcal{U}}$); or select edge $(i, j)$ with probability proportional to $(D_{ii}D_{jj})^{-\frac{1}{2}}$; or according to edge-attribution assigned by PIG (Sundararajan et al., 2017) ($S_{\mathrm{PIG}}$); or using our method Greedy PIG (Alg. 1) ($S_{\mathrm{GreedyPIG}}$), with

$$g^{\mathrm{GNN}}(\mathbf{s}; A^{(0)}, A) = A^{(0)} + \mathbf{s}\odot(A - A^{(0)}), \quad \text{with all-zero } A^{(0)} = \mathbf{0}, \quad \text{and sparse } \mathbf{s} \in \mathbb{R}^{m\times m}. \quad (9)$$

Figure 3 shows that Greedy PIG can better compress the graph while maintaining performance of the underlying model. The GCN model was pre-trained on the full graph with TF-GNN (Ferludin et al., 2022), on datasets of **ogbn-arxiv** from Hu et al. (2020), and **Cora** and **Pubmed** from (Yang et al., 2016). Inference on the full graph (using all edges) gives performance indicated with a dashed-red line. We observe that removing $> 50\%$ of the graph edges, using GreedyPIG, has negligible effect on GCN performance. For the GCN model assumptions, we ensure the selected edges $S \subseteq \mathcal{E}$ correspond to symmetric $A_S = A_S^\top$ by projecting the output of the gradient oracle on to the space of symmetric matrices—by averaging the (sparse) gradient with its transpose, *i.e.*, by re-defining:

$$\widetilde{\nabla_\mathbf{s} g(\mathbf{s})} \triangleq \frac{1}{2}\frac{\partial}{\partial\mathbf{s}}g^{\mathrm{GNN}}(\mathbf{s}) + \frac{1}{2}\left(\frac{\partial}{\partial\mathbf{s}}g^{\mathrm{GNN}}(\mathbf{s})\right)^\top \quad \text{for use in Alg. 1} \quad (10)$$

We also use Greedy PIG to interpret the GCN. Specifically, zooming-into any particular graph node, we would like to to explain the subgraph around the node that leads to the GCN output. Due to space constraints, we report in the appendix subgraphs around nodes in the ogbn-arxiv dataset. Note that Sanchez-Lengeling et al. (2020) apply integrated gradients on smaller graphs (e.g., chemical molecules with at most 100 edges), whereas we consider large graphs (e.g., millions of edges).

### 5.3    POST-HOC FEATURE SELECTION ON CRITEO TABULAR CTR DATASET

Feature selection is the task of selecting a subset of features on which to train a model, and dropping the remaining features. There are many approaches for selecting the most predictive set of features (Li et al., 2017). Even though in general feature selection and model training can be intertwined, often it is easier to decouple these processes. Specifically, given a trained model, it is often desirable to perform *post-hoc* feature selection, where one only has the ability to inspect and not modify the model. This is also related to global feature attribution (Zhang et al., 2021).

**Methodology.** In Table 1, we compare the quality of different attribution methods for post-hoc feature selection. To evaluate the quality of feature selection, we employ the following approach: **1.** Train a neural network model using all features. **2.** Run each attribution method to compute *global* attribution scores which are aggregate attribution scores over many input examples. **3.** Pick the top-$k$ attributed features, and train new pruned models that only use these top features. Report the validation loss of the resulting pruned models for different values of $k$. The results of the methodology described above applied on a random subset of the Criteo dataset are presented in Table 1. We follow the setup of Yasuda et al. (2023), who define a dense neural network with three hidden layers with sizes 768, 256 and 128. To make the comparison fair, we guarantee that all algorithms consume the same amount of data (gradient batches). For example, Greedy PIG with $T = 1$ steps (which we also call Sequential Gradient, see Section 3.5), uses 5x more data batches for each gradient computation than Greedy PIG with $T = 5$.

Table 1: Cross-entropy loss on Criteo CTR dataset using only the top-$k$ features. Reported is the minimum validation loss after 20K training steps with batch size 512. During selection, all of the algorithms consume the same number of batches.

| Algorithm | $k = 5$ | $k = 10$ | $k = 20$ | $k = 30$ |
|---|---|---|---|---|
| Integrated Gradients ($T = 39$) | 0.4827 | 0.4641 | 0.4605 | 0.4533 |
| Greedy PIG ($T = 1$) | 0.4728 | 0.4629 | 0.4551 | **0.4508** |
| Greedy PIG ($T = 5$) | **0.4723** | **0.4627** | **0.4531** | 0.4509 |

## 6 RELATED WORK

Feature attribution, also known as computing the *saliency* of each feature consists of a large class of methods for explaining the predictions of neural networks. The attribution describes the impact of each feature on the neural network's output. In the following, we start with most-related to our research, and then move onto broader and further topics.

**Integrated gradients.** Our work is most related to, and generalizes to set functions the work of Sundararajan et al. (2017); Kapishnikov et al. (2021); Lundstrom et al. (2022); Qi et al. (2019); Goh et al. (2021); Sattarzadeh et al. (2021). In their seminal work, Sundararajan et al. (2017) proposed path integrated gradients (PIG), a method that assigns an importance score to each feature by integrating the partial derivative with respect to a feature along a path that interpolates between the background "*baseline*" and the input features. PIG, however, has some limitations. For example, it can be sensitive to the choice of baseline input. Further, it can be difficult to interpret the results of path integrated gradients when there are duplicate features or features that carry common information. It has also been observed that PIG is sensitive to input noise (Smilkov et al., 2017), model re-trains (Hooker et al., 2019), and in some cases provably unable to identify spurious features (Bilodeau et al., 2022).

Further discussion of related work is deferred to Appendix A.

## 7 CONCLUSION

We view feature attribution and feature selection as instances of subset selection, which allows us to apply well-established theories and approximations from the field of discrete and submodular optimization. Through these, we give a greedy approximation using path integrated gradients (PIG), which we coin Greedy PIG. We show that Greedy PIG can succeed in scenarios where other integrated gradient methods fail, e.g., when features are correlated. We qualitatively show the efficacy of Greedy PIG—it explains predictions of ImageNet-trained model as attributions on input images, and it explains Graph Neural Network (GNN) as attributions on edges on ogbn-arxiv graph. We quantitatively evaluate Greedy PIG for feature selection on the Criteo CTR problem, and by compressing a graph by preserving only a subset of its edges while maintaining GNN performance. Our evaluations show that Greedy PIG gives qualitatively better model interpretations than alternatives, and scores higher on quantitative evaluation metrics.

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
