APPENDIX

## A    RELATED WORK

Simonyan et al. (2014) used gradient ascent to explore the internal workings of a neural network and to quantify the effect of each individual pixel on the classification prediction. Springenberg et al. (2015) presented a neural network that is composed entirely of convolutional layers and introduced guided backpropagation, a technique for eliminating negative signals generated during backpropagation when performing standard gradient ascent. Grad-CAM (Selvaraju et al., 2017) produces a saliency map by using the gradients of any target concept flowing into the final convolutional layer. SmoothGrad (Smilkov et al., 2017) starts by computing the gradient of the class score function with respect to the input image. However, it visually sharpens the gradient-based sensitivity maps by adding noise to the input image and computing the gradient for each of these perturbed versions of the image. Averaging the sensitivity maps together produces a clearer result. For a comprehensive survey of feature attribution and neural network interpretability, see Zhang et al. (2021).

**Shapley value.**    Shapley value was originally introduced in game theory (Shapley, 1953). While it can be directly applied to explain the predictions of neural networks, it requires extremely high computational complexity. Lundberg & Lee (2017) proposed the SHapley Additive exPlanations (SHAP) algorithm, which approximates the Shapley value of features. Sundararajan & Najmi (2020) employed an axiomatic approach to investigate the differences between various versions of the Shapley value for attribution, and they also discussed a technique called Baseline Shapley (BShap). The following are other Shapley value-based methods (Chen et al., 2019; Frye et al., 2021).

**Gradient and backpropgation.**    Another class of interpretability methods is based on gradient and backpropagation. These methods compute the gradient of each feature, which is used as a measure of the feature's importance (Baehrens et al., 2010; Simonyan et al., 2014). Integrated gradient is an instance of the gradient-based method. Backpropagation-based methods design backpropagation rules for convolutional, pooling, and nonlinear activation layers so they can assign importance scores in a fair and reasonable manner during backpropagation (Shrikumar et al., 2017). The guided backpropagation (GBP) algorithm (Springenberg et al., 2015), the layer-wise relevance propagation (LRP) algorithm (Bach et al., 2015), and integrated gradient (IG) algorithm (Sundararajan et al., 2017) are three notable examples of this class of methods.

**Model-agnostic explanation.**    Ribeiro et al. (2016) proposed Local Interpretable Model-Agnostic Explanations (LIME), a method that uses a trained local proxy model to provide explanation results without the need for backpropagation to compute gradients. Plumb et al. (2018) employed a similar method that relies on a local linear model for explanation.

## B    MISSING PROOFS

### B.1    PROOF OF LEMMA 4.3

*Proof.* By Taylor's theorem, we have

$$\nabla g(\boldsymbol{w}) = \nabla g(\boldsymbol{w}_{\{i\}}) + \overline{\boldsymbol{H}} \boldsymbol{w}_{[n]\setminus\{i\}},$$

where $\overline{\boldsymbol{H}}$ is an average Hessian on the path from $\boldsymbol{w}$ to $\boldsymbol{w}_{\{i\}}$. Then,

$$\int_{t=0}^{1} \nabla_i g(t\mathbf{1}) dt = \int_{t=0}^{1} \nabla_i g(t\mathbf{1}_i) dt + \int_{t=0}^{1} \mathbf{1}_i^\top \overline{\boldsymbol{H}}(t) t(\mathbf{1}_{[n]\setminus\{i\}}) dt$$

Now, we know that $\left| \mathbf{1}_i^\top \overline{\boldsymbol{H}}(t)(\mathbf{1}_{[n]\setminus\{i\}}) \right| \le K$, and $\int_{t=0}^{1} \nabla_i g(t\mathbf{1}_i) dt = g(\mathbf{1}_i) - g(\mathbf{0})$, so

$$\left| \int_{t=0}^{1} \nabla_i g(t\mathbf{1}) dt - (g(\mathbf{1}_i) - g(\mathbf{0})) \right| \le K/2 \,.$$

However, by the non-correlation property, this implies that

$$\nabla_S g(\boldsymbol{w}) = \nabla_S g(\boldsymbol{w}_S)$$

As a result, we have that

$$
\begin{aligned}
\langle \mathbf{1}, \boldsymbol{a}_S \rangle &= \langle \mathbf{1}, -\int_{t=0}^1 \nabla_S g(t\mathbf{1}) dt \rangle \\
&= \langle \mathbf{1}, -\int_{t=0}^1 \nabla_S g(t\mathbf{1}_S) dt \rangle \\
&= g(\mathbf{0}) - g(\mathbf{1}_S),
\end{aligned}
$$

which completes the proof. $\qquad\square$

### B.2 INTEGRATED GRADIENTS FOR LINEAR REGRESSION

**Lemma B.1.** *We consider the function $g(\boldsymbol{w}) = -\|\boldsymbol{A}(\boldsymbol{x} \odot \boldsymbol{w}) - \boldsymbol{b}\|^2$, where $\boldsymbol{w} \in [0,1]^n$, $\boldsymbol{A} \in \mathbb{R}^{m \times n}$, $\boldsymbol{b} \in \mathbb{R}^m$ and $\boldsymbol{x}$ is the optimal solution of the linear regression problem $\min_{\boldsymbol{x}} \|\boldsymbol{A}\boldsymbol{x} - \boldsymbol{b}\|^2$. Then, the integrated gradient scores are given by*

$$
\int_{t=0}^1 \nabla g(t\mathbf{1}) dt = \boldsymbol{x} \odot \nabla g(\mathbf{0}).
$$

*Proof.* We note that, for any $t \in [0,1]$, and using the known fact that $\boldsymbol{x} = (\boldsymbol{A}^\top \boldsymbol{A})^+ \boldsymbol{A}^\top \boldsymbol{b}$, we have

$$
\begin{aligned}
\nabla g(t\mathbf{1}) &= 2\boldsymbol{X}^\top \boldsymbol{A}^\top (\boldsymbol{A} t \boldsymbol{x} - \boldsymbol{b}) \\
&= 2t\boldsymbol{X}^\top \boldsymbol{A}^\top \boldsymbol{b} - 2\boldsymbol{A}^\top \boldsymbol{b} \\
&= -2(1-t)\boldsymbol{X}^\top \boldsymbol{A}^\top \boldsymbol{b} \\
&= 2(1-t)\boldsymbol{x} \odot \nabla g(\mathbf{0}),
\end{aligned}
$$

where $\boldsymbol{X} = \mathrm{diag}(\boldsymbol{x})$. Then, we conclude that

$$
\begin{aligned}
\int_{t=0}^1 \nabla g(t\mathbf{1}) dt &= 2\boldsymbol{x} \odot \nabla g(\mathbf{0}) \int_{t=0}^1 (1-t) dt \\
&= \boldsymbol{x} \odot \nabla g(\mathbf{0}). \qquad\square
\end{aligned}
$$

## C EXPERIMENTAL DETAILS AND MORE EXPERIMENTS

### C.1 EXPLAINING GNN PREDICTIONS

For these experiments, we used GCN model pretrained on ogbn-arxiv, circa §5.2. In these experiments, we want to select graph edges that explain classification of one node (contrast to graph compression §5.2 where we select edges that maintain classification of *all* nodes). Given node node $i \in [m]$, we want to select neighbors of $i$, as well as their neighbors, and their neighbors, ..., up-to the depth of the trained GCN (we used 3 GCN layers), that would make the GCN model prediction unchanged as compared to the full subgraph around node $i$. The gradient orcale was modified to only return nonzero gradients to edges connecting already-discovered nodes.

Figure 4 shows a qualitative evaluation of GreedyPIG. One could argue: an article should only cite another if it is related. However, the degree of relatedness can vary. When explaining GNN predictions, we hope the explanation method to select a subgraph that is very related to the center node. The figure shows that random edges around a center node can be less-related to the center node, than if the edges were chosen using GreedyPIG. In the top row of Fig. 4, MixHop paper and blue neighbors are related to GNNs and message passing (MP), whereas red nodes include embedding methods (not MP), or non-GNN applications. In the bottom row, we see that the random edge selection quickly diverged to articles within the NLP domain and otherwise unrelated applications of DeepWalk.

Finally, we shortcut the paper names to reduce visual clutter. For completeness, the full titles of the papers, as appearing in ogbn-arxiv dataset (Hu et al., 2020), are as follows.

- **DeepWalk as Factorization**: comprehend deepwalk as matrix factorization.

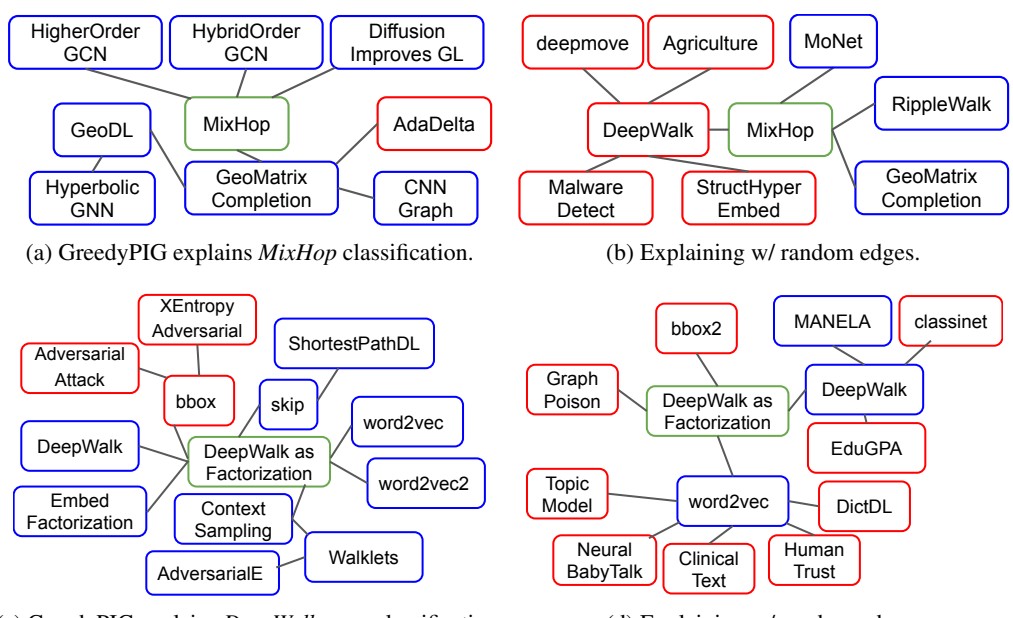

(a) GreedyPIG explains *MixHop* classification.

(b) Explaining w/ random edges.

(c) GreedyPIG explains *DeepWalk as* ... classification.

(d) Explaining w/ random edges.

Figure 4: Pre-trained Graph Neural Network (GNN) is asked to predict article category of center node (green). GreedyPIG (left column) was used to select subgraph around center node that maintains the GNN prediction and we compared it to selecting random adjacent edges. We manually labeled the selected nodes as *strongly-related* (blue) and *less-related* (red).

- **MixHop**: mixhop higher order graph convolutional architectures via sparsified neighborhood mixing.

- **DeepWalk**: deepwalk online learning of social representations.

- **Context Sampling**: vertex context sampling for weighted network embedding

- **word2vec**: efficient estimation of word representations in vector space

- **word2vec2**: distributed representations of words and phrases and their compositionality

- **skip**: don t walk skip online learning of multi scale network embeddings

- **bbox**: a restricted black box adversarial framework towards attacking graph embedding models

- **AdversarialAttack**: threat of adversarial attacks on deep learning in computer vision a survey

- **ShortestPathDL**: shortest path distance approximation using deep learning techniques.

- **Walklets**: walklets multiscale graph embeddings for interpretable network classification

- **AdversarialE**: learning graph embedding with adversarial training methods

- **Embed Factorization**: network embedding as matrix factorization unifying deepwalk line pte and node2vec.

- **bbox**: a restricted black box adversarial framework towards attacking graph embedding models

- **XEntropyAdvers**: cross entropy loss and low rank features have responsibility for adversarial examples"

- **bbox2**: the general black box attack method for graph neural networks

- **HumanTrust**: the transfer of human trust in robot capabilities across tasks.

- **Neural BabyTalk**: neural baby talk.

- **DictDL**: integrating dictionary feature into a deep learning model for disease named entity recognition.

- **MANELA**: manela a multi agent algorithm for learning network embeddings.

- **ClinicalText**: clinical text generation through leveraging medical concept and relations.

- **EduGPA**: will this course increase or decrease your gpa towards grade aware course recommendation.

- **GeoMatrixCompletion**: convolutional geometric matrix completion.

- **GeoDL**: geometric deep learning going beyond euclidean data.

- **AdaDelta**: adadelta an adaptive learning rate method.

- **HigherOrderGCN**: higher order weighted graph convolutional networks.

- **HybridOrderGCN**: hybrid low order and higher order graph convolutional networks.

- **CNNGraph**: deep convolutional networks on graph structured data.

- **HyperbolicGNN**: hyperbolic graph neural networks.

- **deepmove**: deepmove learning place representations through large scale movement data.

- **Agriculture**: cultivating online question routing in a question and answering community for agriculture.

- **RippleWalk**: ripple walk training a subgraph based training framework for large and deep graph neural network.

- **MalwareDetect**: aidroid when heterogeneous information network marries deep neural network for real time android malware detection.

- **MoNet**: monet debiasing graph embeddings via the metadata orthogonal training unit.

- **StructHyperEmbed**: structural deep embedding for hyper networks.

## C.2   Feature attribution on images

For the experiments in Section 5.1, we used the MobilenetV2 image classification network pretrained on Imagenet, using the implementation from the tf.keras library. To implement the algorithms from previous work that we used in comparisons, i.e. integrated gradients and guided integrated gradients, we used Saliency (2017), which is an collection of implementations of state of the art saliency methods. In order to generate Figure 1, we first took a random sample of Imagenet examples from various classes, and for each sample we first applied the preprocessing routine used in MobilenetV2, which includes centering and resizing to $224 \times 224$ pixels with 3 color channels. This $224 \times 224 \times 3$ tensor of 3 dimensions is the input to the neural network, and as a result it is also the shape of the attribution map.

We ran different attribution algorithms and sorted the absolute values of attribution values. Then, we picked 100 equally spaced values for $k$, from 0 to $224 \cdot 224 \cdot 3$, and generated an image using only the top-$k$ attribution values (and replacing the rest by the baseline, which in this case is a gray image). For guided PIG and SmoothGrad, we used the default settings from the saliency library.

## C.3   Pointing game

In this section, we consider the *pointing game* defined by Bohle et al. (2021); Böhle et al. (2022), which is a sanity check for the usefulness of feature attribution methods on images. In this game, examples of different classes are stitched together in a grid to form a new image. Then, this new image is fed to the attribution method, with the goal to explain one of the four classes. Ideally, the highest attributions should be concentrated in the quadrant that corresponds to the selected class. In Figures 5 and 6, we see one such example that compares the attributions of integrated gradients and Greedy PIG.

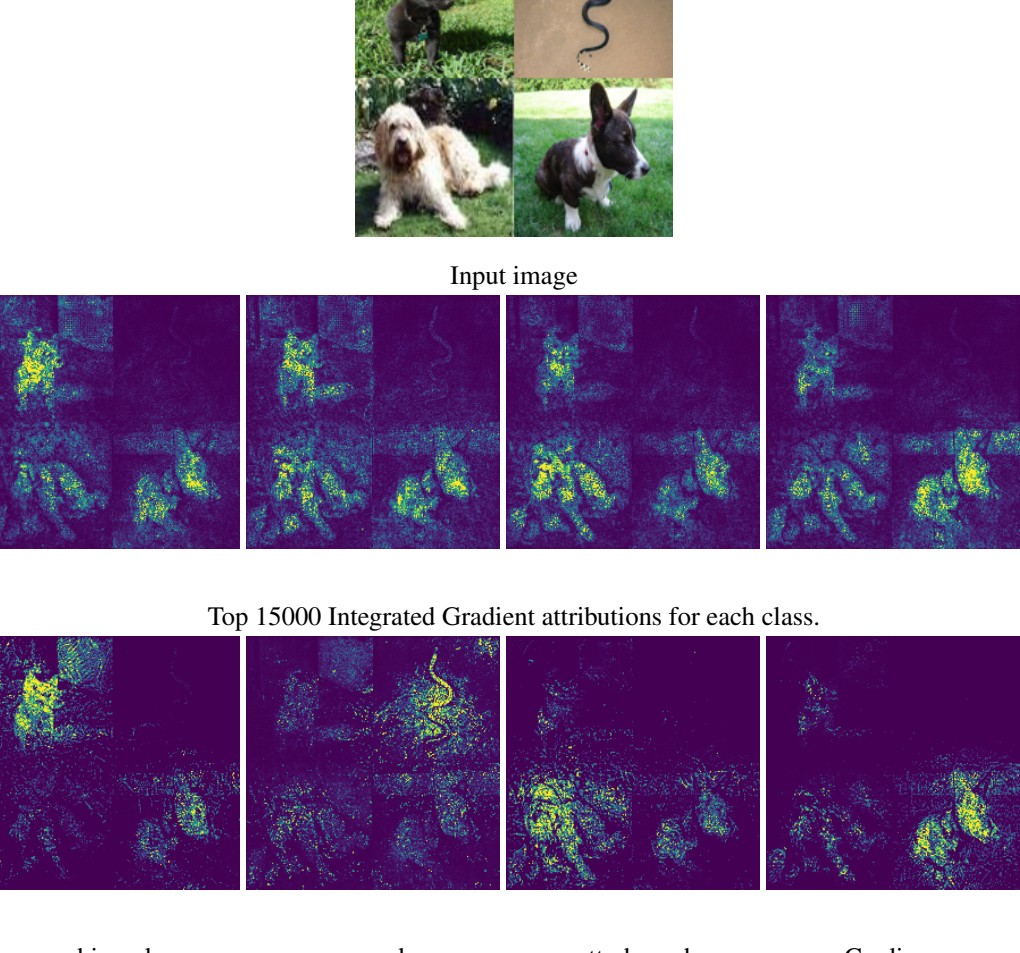

Figure 5: The input image is generated as a 2x2 grid of images from different classes, here schipperke, sea snake, otterhound and Cardigan. Ideally, attributions should be concentrated in the quadrant associated with the respective class.

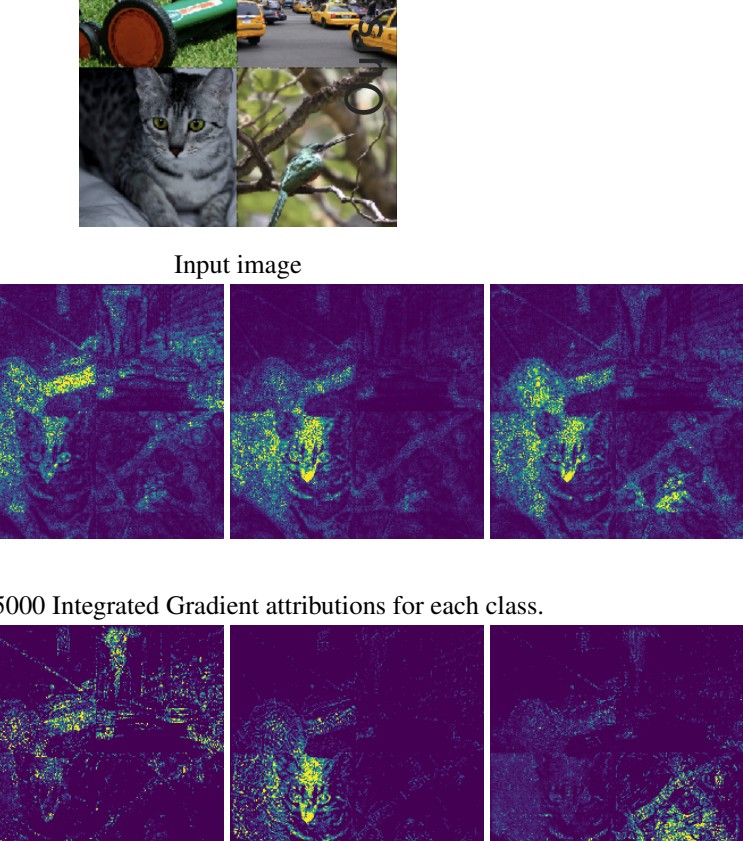

Input image

Top 15000 Integrated Gradient attributions for each class.

lawnmower                cab                Egyptian cat                jacamar

Top 15000 Greedy PIG attributions for each class.

Figure 6: The input image is generated as a 2x2 grid of images from different classes, here lawnmower, cab, Egyptian cat and jacamar. Ideally, attributions should be concentrated in the quadrant associated with the respective class.

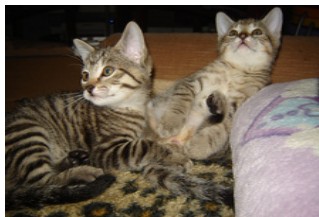 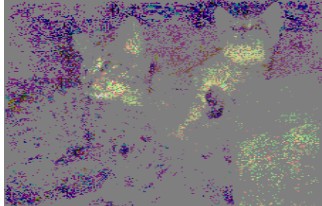 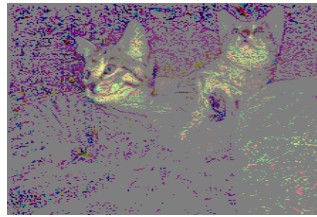

(a) Top 15000 features attributed by integrated gradients (left) vs. GreedyPIG (right). The softmax output for true class "tabby" *after pruning* unselected features is $3 \cdot 10^{-5}$ for integrated gradients vs. 0.9973 for GreedyPIG.

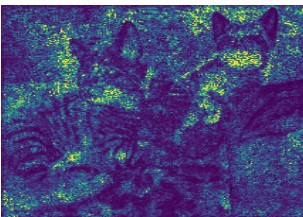 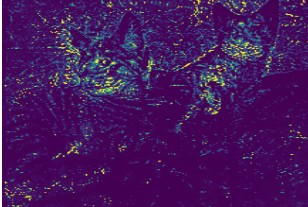

(b) Heatmap of feature attributions by integrated gradients (left) vs. GreedyPIG (right).

Figure 7: Illustration of attribution algorithms on an example from Imagenet labeled "tabby" (left column). Integrated gradients (IG) (middle) ran for 2000 steps, while GreedyPIG (right) ran for 100 rounds each with 20 steps. Both IG and GreedyPIG maximized softmax objective.

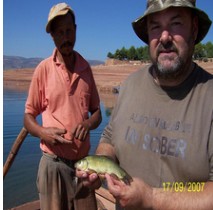 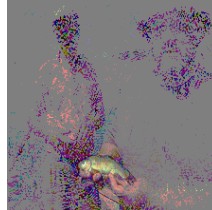 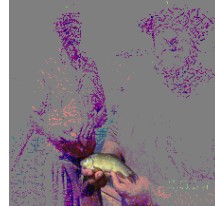

Figure 8: Illustration of different attribution objectives on an example from Imagenet labeled "tench". Left: Original image. Middle: Top 15000 feature attributions by Greedy PIG with the softmax objective, and Right: with the cross-entropy objective.

## C.4 EXAMPLES

## C.5 BLOCK-WISE FEATURE ATTRIBUTION

In the spirit of Kapishnikov et al. (2019), we augment the greedy PIG algorithm with the ability to select patches instead of individual features from a 3-dimensional image tensor. In fact, our implementation allows for specifying arbitrary subset structures that are to be selected as individual features. The results can be found in Figure 9.

To get into more detail, the approach of Kapishnikov et al. (2019) is to first compute attribution scores using integrated gradients, and then iteratively find regions of the image with maximum sum of attributions. The main difference from our approach, is that we invoke the integrated gradient algorithm after *every* block selection, insteado of just once, in line with our adaptive approach. Specifically, for a given block size $b \times b$, after each round of greedy PIG we compute the $b \times b$ patch with the maximum sum of attributions, and select this patch. It should be noted that selected features have a score of 0 in future iterations.

## C.6 FEATURE SELECTION

For the experiments in Section 5.3, we used a model identical to Yasuda et al. (2023), which is a 3-hidden layer neural network with ReLU activations. We implemented minibatch versions of integrated gradients and greedy PIG.

**Note:** While the attributions computed by the integrated gradients algorithm can be averaged over the whole dataset to compute global attribution scores, this is not necessarily the case with the greedy PIG attribution scores. This is another place where the formulation in Definition 4.1 will be handy. Specifically, given a model $f(\cdot; \boldsymbol{\theta})$ with parameters $\boldsymbol{\theta}$, data $(\boldsymbol{X}, \boldsymbol{y})$ and an aggregate loss function $\ell$ over the data, we define the post-hoc feature selection problem simply by $G(S) := -\ell(\boldsymbol{y}, f(\boldsymbol{X}_S; \boldsymbol{\theta}))$. Note that this is *not* equivalent to averaging the greedy PIG attributions across the dataset. While a

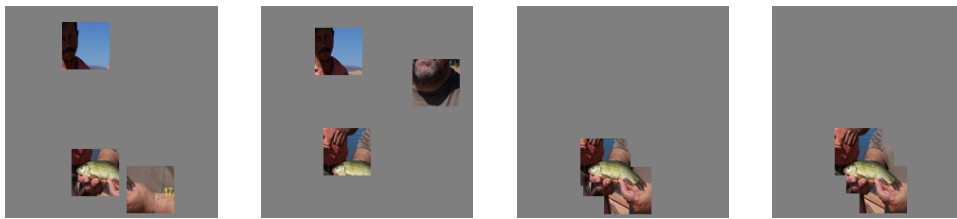

(a) Block-wise attributions. From left to right: Integrated gradients with softmax, integrated gradients with cross entropy, greedy PIG with softmax, greedy PIG with cross entropy.



(b) Block-wise attributions. From left to right: Integrated gradients with softmax, greedy PIG with softmax, integrated gradients with cross entropy, greedy PIG with cross entropy.

Figure 9: Illustration of different attribution objectives on an example from Imagenet labeled "tench", with different block sizes.

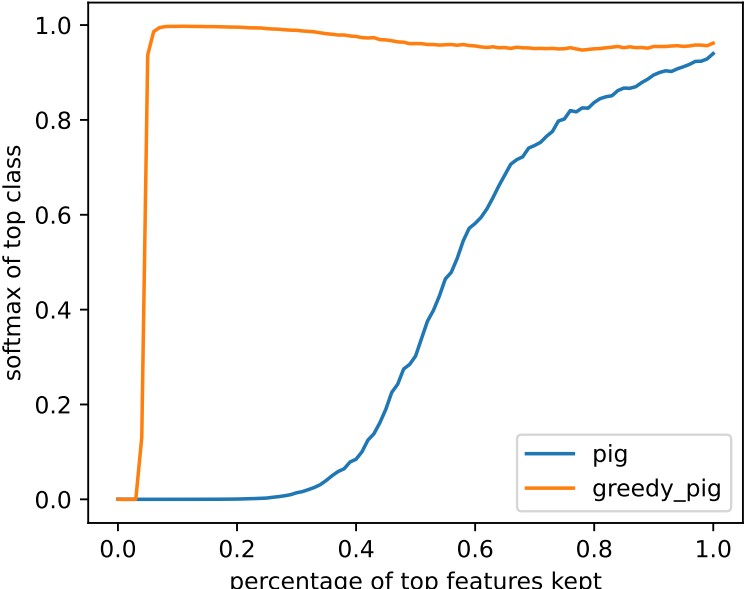

Figure 10: Comparing pig and greedy_pig on imagenet attribution using ResNet50. The AUC scores for pig and greedy_pig are $0.39443$ and $0.9262$ respectively.

naive implementation of Algorithm 1 would require multiple full batch gradient evaluations in each round, we implement a mini-batch version, which instead samples a number of data batches in each round, and computes the gradient over these batches.

Specifically, given a number $n = 150000$ of available data batches, each of size $512$, and a number $g = 39$ of gradients to be evaluated (this is the number of integrated gradient steps), we randomly shuffle the batches and evaluate each of the gradients on a random set of $n/g \approx 3846$ batches (i.e.

the gradient is averaged over all these batches). In this way, we can ensure that all algorithms use the same amount of data, and the algorithms are scalable enough to be used in large scale settings.