# OpenReview forum: "Greedy PIG: Adaptive Integrated Gradients"
_ICLR.cc/2024/Conference — Submitted to ICLR 2024_

### Official Review · Reviewer_WqYq · 2023-10-24

**Soundness:** 3 good
**Presentation:** 4 excellent
**Contribution:** 2 fair
**Rating:** 6
**Confidence:** 3

**Summary:**

This research study bridges the gap between two domains of deep learning: attribution and feature selection. They propose a novel unified theoretical framework. The resulting method, although similar to previous work, uses feature selection in order to increase the robustness of the attribution evaluation. Their result show that the proposed Greedy PIG vastly outperforms some previous methods in terms of Softmax AUC and KL divergence AUC.

**Strengths:**

In my opinion, explainability and compression are of paramount importance in deep learning. In this paper, the authors show a limitation of existing methods. As a result, Greedy PIG is specifically designed to mitigate this issue and achieves remarkable results.

**Weaknesses:**

I have three concerns with this work as it stands.
1. The method is designed to perform well when evaluated using the Softmax AUC which is not the most commonly used metric (insertion and deletion scores are). How the Greedy PIG compare with other methods using these metrics?
2. A recent method IDGI [1] was introduced
3. Although ConvNets are still popular, the study would strongly benefit from an evaluation on Transformers, e.g. ViT.

[1] Yang, Ruo, Binghui Wang, and Mustafa Bilgic. "IDGI: A Framework to Eliminate Explanation Noise from Integrated Gradients." Proceedings of the IEEE/CVF Conference on Computer Vision and Pattern Recognition. 2023.

**Questions:**

On top of my previous concerns, I would like to ask if the authors could the authors share their code (at least on an example). I am intrigued by the difference in performance with GIG which in my understanding is very similar to the proposed method

---

> ### Author Response · Authors · 2023-11-20
> **Response to Reviewer WqYq**
>
> We thank the reviewer for the valuable comments and for bringing some related work to our attention.
>
> >The method is designed to perform well when evaluated using the Softmax AUC which is not the most commonly used metric (insertion and deletion scores are). How the Greedy PIG compare with other methods using these metrics?
>
> After reading through the work of Petsiuk et al. 2018, we conclude that the scores we are using are indeed the insertion scores (the x-axis is the number of selected pixel values and not the entropic quantity used in Kapishnikov et al. 2019). As we were not aware of Petsiuk et al. 2018, we implicitly re-defined insertion scores as a variation of the SIC AUC scores of Kapishnikov et al. 2019 (and still called them SIC AUC scores). We thank the reviewer for pointing us to this work, this simplifies our exposition! In the revised version we will clarify that we are using insertion scores and cite Petsiuk et al. 2018.
>
> > A recent method IDGI [1] was introduced
>
> Thank you for letting us know about this work, indeed it seems very related and interesting. We will read it carefully and cite it in the next version.
>
> >On top of my previous concerns, I would like to ask if the authors could the authors share their code (at least on an example). I am intrigued by the difference in performance with GIG which in my understanding is very similar to the proposed method
>
> Yes, we are preparing to publicly release our code. This requires some time on our side due to approvals, but we hope to be able to finish it by the revision deadline.
>
> Petsiuk, Vitali, Abir Das, and Kate Saenko. "Rise: Randomized input sampling for explanation of black-box models." arXiv preprint arXiv:1806.07421 (2018).

---

> > ### Author Response · Authors · 2023-11-21
> > **Update on code**
> >
> > We have now uploaded a colab with the full code of Greedy PIG and an example here: https://colab.research.google.com/drive/16fV4gAyim-K5riqlhPyxY_kqIIpGqGik

---

### Official Review · Reviewer_mYhW · 2023-10-31

**Soundness:** 2 fair
**Presentation:** 2 fair
**Contribution:** 3 good
**Rating:** 3
**Confidence:** 4

**Summary:**

The paper introduces an improvement over Integrated gradients by advocating to make it adaptive. They do so by recursively taking the top-k attribution features, adding it to the current baseline, and recomputing the path gradients. The authors then show that their attribution method outperforms previous modifications to integrated gradients on several performance AUC metrics.

**Strengths:**

1. I like the idea of adaptively choosing the baseline in order to break the redundancies between features involved. However, I think this aspect of the paper has not been properly evaluated by the authors. I expand on this in the weakness section.

2. The proposed modification to integrated gradients outperforms previous methods in literature in AUC curves which show that their method chooses features that are more important for prediction than other attribution methods.

**Weaknesses:**

The motivation of this work is not adequately backed up with theory or experiments. Moreover the writing is weak making the paper hard to read. I would expand on this in the following points.

1. The stated motivation for greedy PIG is to make the attributions more robust to feature correlations. However this aspect has never been explicitly evaluated in experiments. Lemma 4.4 is an attempt to theoretically justify why integrated gradients would fail when redundant features are present, however no proof is provided in the paper to evaluate the correctness of the statement. Moreover, it is not clear how greedy PIG solves the issue stated in Lemma 4.4. Clarifying this would further strengthen the motivations of this work.

2. The Proof of Lemma 4.3 is not clear. Why is the hessian bounded by K? What is the non-correlation property of g? What is \bar{H}. The authors say this is average on a path from w to w_{i}. What is the formulae for computing this average? how is the path computed? what is w_{I}. The details should be clarified to the reader.

3. More generally, it is not clear to me what g is in the paper. Is it the neural network function f as in equation 1? Section 3.3 says this is a continuous extension that allows optimization of equation 3, however equation 3 is never optimized in their greedyPIG algorithm.

4. For the experiments, what is the value of z, chosen for the greedy-PIG algorithm in each instance. An ablation study on the effect of z (the number of top-z features selected in each iteration) on the different metrics would be interesting as it would show the robustness of the method on the choice of z. If one would want to break correlations, is the ideal value z=1?

5. It is not clear what is the Sequential Gradient, the authors refer to in this paper. Is it eq (1) evaluated at one single point instead of a discretization on N points? If yes, how is this point selected? how accurate is this estimation?

5. Please describe what the point game is in more detail. I understand it was proposed in an earlier paper, so I recommend this be added to the appendix. Otherwise it is not clear to the reader at all what is been shown. Is the network (that is explained) trained on a new dataset that includes images arranged in a 3x3 grid, or is it through the same network? If yes does it not affect the performance of the original network which was trained on clean imageS?  The statement "We generate 2x2 grids of the highest prediction confidence images, and obtain the attribution results for each class" is unclear. What does highest prediction confidence images mean? How are the attribution results obtained?

**Questions:**

Refer to the weaknesses above.

---

> ### Author Response · Authors · 2023-11-20
> **Response to Reviewer mYhW**
>
> We thank the reviewer for their comments and suggestions.
>
> >The stated motivation for greedy PIG is to make the attributions more robust to feature correlations. However this aspect has never been explicitly evaluated in experiments. Lemma 4.4 is an attempt to theoretically justify why integrated gradients would fail when redundant features are present, however no proof is provided in the paper to evaluate the correctness of the statement. Moreover, it is not clear how greedy PIG solves the issue stated in Lemma 4.4. Clarifying this would further strengthen the motivations of this work.
>
> We will add a proof of Lemma 4.4 in the next version (it is very short). How Greedy PIG avoids the redundancy issue in Lemma 4.4 is explained in the paragraph right after Lemma 4.4. In short, because of adaptivity, selecting a feature $X$ in round $i$ will lead to avoiding redundant features in rounds $> i$, since their PIG score will be $0$. More generally on adaptivity, there are theoretical lower bounds (explained in Section 3.4) showing that even for the special case of submodular function maximization, multiple rounds of adaptive function evaluations are necessary for computing an optimal solution (Balkanski et al. 2018).
>
> >The Proof of Lemma 4.3 is not clear. Why is the hessian bounded by K? What is the non-correlation property of g? What is \bar{H}. The authors say this is average on a path from w to w_{i}. What is the formulae for computing this average? how is the path computed? what is w_{I}. The details should be clarified to the reader.
>
> We thank the reviewer for noticing an editing mistake. Some leftover text from a previously removed lemma was incorrectly pasted into the proof which did not make sense. As for $w_{\{i\}}$, it is a vector that is equal to $w_i$ at $i$ and $0$ everywhere else, as defined in the preliminaries (Section 2.2). The Hessian averaging is a direct consequence of Taylor’s theorem (see e.g. a stronger version in Theorem 1 here: https://www.cs.princeton.edu/courses/archive/fall18/cos597G/lecnotes/lecture3.pdf). We will clarify this better in the revised version.
>
> >More generally, it is not clear to me what g is in the paper. Is it the neural network function f as in equation 1? Section 3.3 says this is a continuous extension that allows optimization of equation 3, however equation 3 is never optimized in their greedyPIG algorithm.
>
> In our work, we seek to solve the subset selection problem (3) by using a continuous relaxation $g$ of $G$, which allows us to compute gradients. In fact it is sometimes easier to directly think in terms of $g$ instead of $G$. $g$ can be any continuous function that matches $G$ on the vertices of the hypercube, but the choice of $g$ is usually natural (e.g., the training loss of a neural network $f(\cdot; \theta)$).
>
> >For the experiments, what is the value of z, chosen for the greedy-PIG algorithm in each instance. An ablation study on the effect of z (the number of top-z features selected in each iteration) on the different metrics would be interesting as it would show the robustness of the method on the choice of z. If one would want to break correlations, is the ideal value z=1?
>
> That’s exactly right, the ideal value for breaking correlations is $z=1$. In all our experiments, we choose $z=k/R$, where $R$ is the number of rounds. This means that $z=1$ implies a large number of rounds, which increases the overall running time (since we need to compute $O(RT)$ gradients, where $T$ is the number of PIG steps) but generally improves the solution quality. We observed that even slightly increasing the number of rounds (e.g. from $R=1$ to $R=5$) gives noticeable improvement in the feature attribution metrics. In the experiments of Figure 1 we picked $R=100$ ($z\approx 1500$) because there ~150k features ($224\times 224\times 3$), and in the experiments of Table 1 we picked $R=k$ ($z=1$) because there are only $39$ features.
>
> >It is not clear what is the Sequential Gradient, the authors refer to in this paper. Is it eq (1) evaluated at one single point instead of a discretization on N points? If yes, how is this point selected? how accurate is this estimation?
>
> Sequential Gradient is Greedy PIG with $T=1$. Specifically, it approximates the integral $\int_{t=0}^1 \nabla g(t {\bf 1}) dt$ by $\nabla g({\bf 0})$. This is accurate if $g$ is close to linear, but based on our results it is still a reliable approach with lower runtime (e.g. see the first Greedy PIG row in Table 1).

---

> > ### Author Response · Authors · 2023-11-20
> > **Response (Part 2)**
> >
> > >Please describe what the point game is in more detail. I understand it was proposed in an earlier paper, so I recommend this be added to the appendix. Otherwise it is not clear to the reader at all what is been shown. Is the network (that is explained) trained on a new dataset that includes images arranged in a 3x3 grid, or is it through the same network? If yes does it not affect the performance of the original network which was trained on clean imageS? The statement "We generate 2x2 grids of the highest prediction confidence images, and obtain the attribution results for each class" is unclear. What does highest prediction confidence images mean? How are the attribution results obtained?
> >
> > In the pointing game, we are given a fixed neural network pre-trained on ImageNet classification. We feed a set of images to it and select those that are classified correctly and with the highest confidence (given by the softmax score of the predicted class). We group these into sets of 4 and arrange these sets into 2x2 grids (the network has not seen these grids before). Then, for each grid and each image in the grid we ask the attribution algorithm to “explain” the class of that image (by finding a small number of input features that maximize the softmax output of that class, as in the experiments in Figure 1). We look at the top ~10% feature attributions and find the quadrant they most frequently reside in. If this is the quadrant that indeed contains the selected image, we count a correct “guess”. Intuitively, this means that the attribution algorithm is able to point to what the neural network is “seeing” to determine its prediction. We will add these clarifications to the revised version.

---

### Official Review · Reviewer_yEX4 · 2023-11-05

**Soundness:** 2 fair
**Presentation:** 2 fair
**Contribution:** 2 fair
**Rating:** 5
**Confidence:** 3

**Summary:**

This paper investigates the problem of feature attribution as an explicit subset selection problem.  Realizing that the main drawback of the path-integrated gradient (PIG) algorithms is their limited ability to handle feature correlations, the authors propose a natural way to account for correlations by a greedy algorithm, i.e., the correlations between already selected variables with the rest of the unselected variables will be eliminated by the greedy selection strategy. Experiments on a wide variety of tasks, including image feature attribution, graph compression/explanation, and the post-hoc feature selection on tabular data demonstrate the effectiveness of the proposed method.

**Strengths:**

1. The authors connect feature attribution and feature selection with a unified discrete optimization framework based on subset selection.
2. Experiments on a wide variety of tasks, including image feature attribution, graph compression/explanation, and the post-hoc feature selection on tabular data demonstrate the effectiveness of the proposed method.

**Weaknesses:**

1. The novelty of the proposed method is limited.  By simply combining feature attribution and feature selection with a unified discrete optimization framework based on subset selection, the authors introduce limited insight into tackling this problem. Equation 7 is a simple extension of Equation 1.
2. The proposed Greedy PIG may introduce a sub-optimal problem.  By greedily selecting the top-attribution features computed by integrated gradients in each round, the proposed method cannot guarantee a global optimal solution for the feature attribution problem. Further, if seeking the global optimal solution for the feature attribution problem is not the goal of this submission, it may be better for the authors to demonstrate that a satisfactory solution will be attained by the proposed method.
3. This paper is not well-written, and more explanation is needed to deeply follow this paper. For example, "feature attribution, the softmax information curve (SIC) of Kapishnikov et al. (2019) can be recovered from (Eq. 3) by setting G(S) to the softmax output of a target class (see Eq. 4)." is quite confused.

**Questions:**

1.  A typo in the second paragraph of the introduction section: "on considers an entire dataset. For literature surveys, see (Zhang et al., 2021) for feature attribution and interpretability see and (Li et al., 2017) for feature selection."

**Details Of Ethics Concerns:**

No ethics review is needed.

---

> ### Author Response · Authors · 2023-11-20
> **Response to Reviewer yEX4**
>
> We thank the reviewer for the comments and for identifying some typos that we will fix in the revised version.
>
> >The novelty of the proposed method is limited. By simply combining feature attribution and feature selection with a unified discrete optimization framework based on subset selection, the authors introduce limited insight into tackling this problem. Equation 7 is a simple extension of Equation 1.
>
> Our contribution is not replacing a set function by a continuous relaxation. Our contribution is introducing **multi-round adaptivity** (inspired by subset selection problems) to solve attribution problems more effectively.
>
> >The proposed Greedy PIG may introduce a sub-optimal problem. By greedily selecting the top-attribution features computed by integrated gradients in each round, the proposed method cannot guarantee a global optimal solution for the feature attribution problem. Further, if seeking the global optimal solution for the feature attribution problem is not the goal of this submission, it may be better for the authors to demonstrate that a satisfactory solution will be attained by the proposed method.
>
> The subset selection problem is NP-hard in general (there are many hardness results for this type of problem, see e.g. Foster et al. 2015). We do not claim to achieve a globally optimal solution but one with good experimental performance in feature attribution tasks. If the reviewer has a suggestion on an algorithm that might return higher quality solutions for the feature attribution objectives, we would be happy to consider it. Further, the greedy algorithm is provably the best achievable algorithm for certain subset selection problems, e.g., monotone submodular maximization subject to a cardinality constraint, unless P = NP.
>
> Foster, Dean, Howard Karloff, and Justin Thaler. "Variable selection is hard." In Conference on Learning Theory, pp. 696-709. PMLR, 2015.

---

> > ### Comment · Reviewer_yEX4 · 2023-12-03
> > **Response to Authors**
> >
> > Thanks for the response from the authors. After reading reviews from other reviewers and the author's response, I still have concerns about the novelty and solidity of the proposed method. Therefore,  I decided to maintain my score.

---

### Official Review · Reviewer_dYDX · 2023-11-08

**Soundness:** 2 fair
**Presentation:** 1 poor
**Contribution:** 2 fair
**Rating:** 3
**Confidence:** 3

**Summary:**

The paper tackles feature attribution, which aims to explain model's decision on an input by assigning to each input feature a score showing their contribution. Different from previous work, the paper proposes to formulate it as a subset selection problem (Sec 2.2 and 3.2), i.e. select the optimal set of features that best explain the model's decision. Inspired by Path Integrated Gradients (PIG), the paper relaxes the objective set function to a continuous function on a path in the hypercube. The problem is then solved using Greedy PIG, an application of PIG in multiple rounds which selects a batch of features at a time to add to the optimal set.

The paper shows good performance compared to PIG-based baselines on feature attribution, GNN compression and feature selection on tabular data.

**Strengths:**

Explainability of deep neural networks is an important topic and the paper tackles an important task toward this goal. Casting feature attribution as subset selection is reasonable.

The paper rightly points out that the correlation of features could lead to wrong attribution. The proposed Greedy PIG algorithm to address this issue seems to result in better performance than the baselines.

**Weaknesses:**

The link between subset selection formulation and Greedy PIG seems very weak. The path going from the formulation to the algorithm should be better clarified. In particular:
  - Why does Greedy PID maximize the objective function? The paper claims that formulating feature attribution as an optimization problem has advantages. But the proposed algorithm seems to be an extension of PIG and has nothing to do with maximizing the real object function.
  - Is the continuous objective function a submodular function? The paper seems to lean a lot on the submodularity of set functions to argue for the approximate optimality of Greedy PIG.

The part of  why Greedy eliminates the effect of feature correlation needs clarification. Is there some mathematical evidence to support claims in paragraph "Why Greedy captures correlations"?

The analysis in Sec 4.2 needs clarification
  - Why is it good that attributions correlate with marginal gains at S=0? If marginal gains are what we want, why don't we directly use them?
  - The paper suggests that H_ij reflects the correlation between features i and j. Is ther any justification?
  - Lemme 4.4 needs a short proof. Also, it considers a very particular form of "feature redundancy". Is this kind of feature redundancy common in practice?

In general, the paper's writing needs major improvements.

**Questions:**

How does the performance depend on parameter z in Algorithm 1?
Function g in Eq. 7 is a typo? Another function g is mentioned earlier in Sec 3.3.

---

> ### Author Response · Authors · 2023-11-20
> **Response to reviewer dYDX**
>
> We thank the reviewer for the very detailed comments. We provide answers to all their questions below.
>
> >Why does Greedy PID maximize the objective function? The paper claims that formulating feature attribution as an optimization problem has advantages. But the proposed algorithm seems to be an extension of PIG and has nothing to do with maximizing the real object function.
>
> The main ingredient of our algorithm is a subroutine that selects the maximum entries of $\nabla g(w)$. These are the entries that (locally) increase the value of $g$ the most, and so each iteration of the Greedy PIG algorithm (for $T=1$ discretization steps) is exactly making a (sparse) step towards maximizing the function (this also holds for $T>1$ discretization steps under assumptions, e.g., if we assume $g$ is a quadratic function). Furthermore, as we show in Lemma 4.3, the PIG scores approximate the marginal gains under some conditions, and so the set of top attributed features will have high marginal gains in some approximate sense. Repeating this process adaptively over multiple rounds significantly boosts the quality of the solution for the maximization problem. This is the intuition why this algorithm performs well for the maximization problem, which is confirmed in the experiments.
>
> While we agree that there is room for further analysis in the theoretical results (which is partially attributable to the PIG algorithm not having strong theoretical backing but a large body of experimental work), we believe that the strong experimental improvements presented in this paper will motivate theoretical study into the properties of the PIG algorithm. That said, our approach of introducing adaptivity is independent of PIG and can be used to boost any algorithm that approximates the marginal gains, so it should be of independent interest.
>
> >Is the continuous objective function a submodular function? The paper seems to lean a lot on the submodularity of set functions to argue for the approximate optimality of Greedy PIG.
>
> Assuming that $G$ is submodular is indeed too strong and generally not met in practice. This is why we do not assume submodularity in our work and the set functions used in the experiments are generally not submodular. That said, based on the experiments many of these problems do exhibit some weaker form of diminishing returns property (see e.g. Figures 1 and 3). Roughly, to get correct classification, knowing the first few features (or edges) gives incrementally a better classification accuracy than incrementally uncovering more features.
>
> > The part of why Greedy eliminates the effect of feature correlation needs clarification. Is there some mathematical evidence to support claims in paragraph "Why Greedy captures correlations"?
>
> Consider features numbered $1,\dots, n$. If the function $G$ is non-decomposable (which is what we mean by correlations between the features), then the total gain of adding all the features, $G(\\{1,\dots,n\\} )-G(\emptyset)$, can be very different from the sum of their marginal gains $G(\\{1\\}) - G(\emptyset) + \dots + G(\\{n\\}) - G(\emptyset)$. Therefore, inserting many of these features simultaneously based on their marginal gains can lead to a suboptimal selection. However, if we add them one at a time, then the sum of the marginal gains of the added features (at the time they were added) are $[G(\\{1\\}) - G(\emptyset)] + [G(\\{1, 2\\}) - G(\\{1\\})] + \dots + [G(\\{1,\dots,n\\}) - G(\\{1,\dots,n-1\\})]$, which is equal to the total gain. This is because upon selecting feature $i$, its correlations with features $>i$ are eliminated, since $i$ is now selected.
>
> > Why is it good that attributions correlate with marginal gains at S=0? If marginal gains are what we want, why don't we directly use them?
>
> The feature with the highest marginal gain is the best greedy choice of feature to select to $S$ for maximizing $G$. If the attributions correlate with the marginal gains, then they can be used as proxies for the marginal gains and selecting the feature with the top attribution will be a high-quality choice of a feature to select, in terms of maximizing $G$. We don't directly use marginal gains because it is computationally inefficient. Computing all marginal gains of $G$ for each selection step requires $O(n)$ evaluations of $G$, whereas PIG only requires evaluating $\nabla g$ a constant number of times.

---

> > ### Author Response · Authors · 2023-11-20
> > **Response (Part 2)**
> >
> > >The paper suggests that H_ij reflects the correlation between features i and j. Is ther any justification?
> >
> > $H_{ij}$ is a measure of correlation between the variables $i, j$ in the context of $g$. If $H_{ij}(w)=0$ for all $w$, then the two features i, j are independent in the sense that $\nabla_i g(...,w_i, …, w_j,...) = \nabla_i g(...,w_i, …,w_j’,...)$ for all $w_i, w_j, w_j’$. This means that selecting variable $j$ (setting $w_j=1$) has no impact on the marginal gain of $i$, and so these two variables can be independently selected. On the other hand if $H_{ij} \neq 0$ then the marginal gain of $i$ is dependent on that of $j$, and so the two features are correlated and selecting $j$ might completely change the marginal gain of $i$ (for example, if $i$ and $j$ are mutually redundant features). A one-shot method cannot capture these effects, which is exactly why we made Greedy PIG adaptive.
> >
> > >Lemme 4.4 needs a short proof. Also, it considers a very particular form of "feature redundancy". Is this kind of feature redundancy common in practice?
> >
> > This kind of redundancy is met e.g. in max pooling operations in neural networks. We will add a short proof in the revised version.
> >
> > >How does the performance depend on parameter z in Algorithm 1?
> >
> > Good question. In our experiments, when we select $k$ features over $R$ rounds we set $z=k/R$. The value of $R$ gives a cost/quality tradeoff: Increasing $R$ generally improves quality (by making the algorithm more adaptive to previous choices) but worsens runtime efficiency.While this per-round batching method is explored somewhat theoretically in “Lazier than lazy greedy” (Mirzasoleiman et al., AAAI 2015), we observed that even slightly increasing the number of rounds (e.g. from $R=1$ to $R=5$) gives noticeable improvement in the feature attribution metrics. In the experiments of Figure 1 we picked $R=100$ ($z\approx 1500$) since there are ~150k features, and in the experiments of Table 1 we picked $R=k$ ($z=1$) since there are only $39$ features.
> >
> > >Function g in Eq. 7 is a typo? Another function g is mentioned earlier in Sec 3.3.
> >
> > This is a typo indeed ($g$ is a relaxation of $G$), thank you for the catch.

---

> ### Comment · Reviewer_dYDX · 2023-11-22
> **Response to authors' rebuttal**
>
> I would like to thank the authors for addressing my questions and concerns. The main issue of this submission is the disconnection between the problem formulation and its proposed solution. Although the solution seems to lead to good results, this issue makes the formulation, which consists of a large part of the paper's content as well as a major claimed contribution, unnecessary. I am not convinced by the authors' rebuttal regarding this point:
>   - The algorithm computes the integration of $\nabla g$, not $\nabla g$, why does selecting the largest entries in this vector lead to a better value for g? The analysis regarding the relation between PIG scores and marginal gains in Lemme 4.3 only apply when the base set S is empty, this analysis does not hold when some points are already selected. I think proving that the algorithm gradually increases the objective function is critical. The paper and the rebuttal have not provided such proof.
>   - The paper is greatly inspired and mentions a lot about literature on submodular optimization but its objective function is not a submocular function (or not proven so). This makes the support for the proposed adaptivity weak and the mention of submodular optimization unnecessary.
>   - The answer for my question on the paragraph "Why Greedy captures correlations" is not clear.
>   - About marginal gains, the point seems to be that approximating marginal gains is good. Even though, computing them could be inefficient, it should be feasible. In order to demonstrate this point, I think a comparison between Greedy PIG and a baseline that uses marginal gains is important.
>
> With the reasons above, I want to keep my initial rating.

---

> > ### Author Response · Authors · 2023-11-22
> > **Authors response**
> >
> > >The algorithm computes the integration of $\nabla g$, not $\nabla g$, why does selecting the largest entries in this vector lead to a better value for g?
> >
> > $\int_{t=0}^1 \nabla g(t{\bf w}) dt$ is equal to $0.5\nabla g({\bf 0})$ for quadratic $g$ optimized at ${\bf 1}$. So for this large family of functions, these two things are the same. Also, the reason why looking at the integral instead of the gradient makes sense for neural networks, is that the gradient is often saturated at ${\bf 0}$. This is the reason why people use the integral instead of the gradient.
> >
> > >The analysis regarding the relation between PIG scores and marginal gains in Lemme 4.3 only apply when the base set S is empty, this analysis does not hold when some points are already selected. I think proving that the algorithm gradually increases the objective function is critical. The paper and the rebuttal have not provided such proof.
> >
> > The reviewer is incorrect. Lemma 4.3 straightforwardly applies to the case $S\neq \emptyset$ by replacing the starting point ${\bf 0}$ by ${\bf 1}_S$.
> >
> > >The paper is greatly inspired and mentions a lot about literature on submodular optimization but its objective function is not a submocular function (or not proven so). This makes the support for the proposed adaptivity weak and the mention of submodular optimization unnecessary.
> >
> > Submodular functions are a special family of set functions. If multi-round adaptivity is important for submodular optimization, it is also important for optimizing more general set families. In fact, we don't understand what the reviewer's concern is here. The experiments clearly show that adaptivity **is** beneficial for our objective functions, even though they are not necessarily submodular.
> >
> > >The answer for my question on the paragraph "Why Greedy captures correlations" is not clear.
> >
> > We are not sure how to respond to this sentence since it does not provide a specific criticism.

---

### Meta-Review · Area_Chair_SoLG · 2023-12-08

**Metareview:**

The paper proposes a new approach to feature attribution by formulating it as a subset selection problem and using Greedy PIG, an application of Path Integrated Gradients, to solve it. As stated in the reviews, the motivation of this work is not aligned with empirical or theoretical evidence. Moreover the writing could largely be improved : the paper is hard to read in its curent form. I recommend the authors to take the feedback from the reviews into account and submit this work to another venue.

**Justification For Why Not Higher Score:**

See meta review.

**Justification For Why Not Lower Score:**

N/A

---

### Decision · Program_Chairs · 2024-01-16

Reject